# Vitexin Mitigates Haloperidol-Induced Orofacial Dyskinesia in Rats through Activation of the Nrf2 Pathway

**DOI:** 10.3390/ijms251810206

**Published:** 2024-09-23

**Authors:** Shu-Mei Chen, Mao-Hsien Wang, Kuo-Chi Chang, Chih-Hsiang Fang, Yi-Wen Lin, Hsiang-Chien Tseng

**Affiliations:** 1Department of Neurosurgery, Taipei Medical University Hospital, Taipei Medical University, Taipei 110, Taiwan; nschenis@gmail.com; 2Department of Surgery, School of Medicine, Taipei Medical University, Taipei 110, Taiwan; 3Department of Anesthesia, En Chu Kon Hospital, Sanshia District, New Taipei City 23702, Taiwan; momonini@me.com; 4Institute of Taiwan Instrument Research, National Applied Research Laboratories, Hsinchu 300092, Taiwan; steven580704@yahoo.com.tw; 5Department of Chemical Engineering and Biotechnology, National Taipei University of Technology, Taipei 10608, Taiwan; 6Department of Orthopedics, College of Medicine, Taipei Medical University, Taipei 110, Taiwan; danny07291991@hotmail.com; 7Institute of Biomedical Engineering, National Taiwan University, Taipei 10051, Taiwan; zhew520@gmail.com; 8Department of Anesthesiology, Shin Kong Wu Ho-Su Memorial Hospital, Taipei 11101, Taiwan; 9School of Medicine, Fu Jen Catholic University, New Taipei City 24205, Taiwan

**Keywords:** vitexin, haloperidol, trigonelline, orofacial dyskinesia, striatum

## Abstract

Vitexin (VTX), a C-glycosylated flavone found in various medicinal herbs, is known for its antioxidant, anti-inflammatory, and neuroprotective properties. This study investigated the protective effects of VTX against orofacial dyskinesia (OD) in rats, induced by haloperidol (HPD), along with the neuroprotective mechanisms underlying these effects. OD was induced by administering HPD (1 mg/kg i.p.) to rats for 21 days, which led to an increase in the frequency of vacuous chewing movements (VCMs) and tongue protrusion (TP). VTX (10 and 30 mg/kg) was given intraperitoneally 60 min after each HPD injection during the same period. On the 21st day, following assessments of OD, the rats were sacrificed, and nitrosative and oxidative stress, antioxidant capacity, mitochondrial function, neuroinflammation, and apoptosis markers in the striatum were measured. HPD effectively induced OD, while VTX significantly reduced HPD-induced OD, decreased oxidative stress, enhanced antioxidant capacity, prevented mitochondrial dysfunction, and reduced neuroinflammatory and apoptotic markers in the striatum, and the protective effects of VTX on both behavioral and biochemical aspects of HPD-induced OD were significantly reduced when trigonelline (TGN), an inhibitor of the nuclear factor erythroid-2-related factor 2 (Nrf2)-mediated pathway, was administered. These findings suggest that VTX provides neuroprotection against HPD-induced OD, potentially through the Nrf2 pathway, indicating its potential as a therapeutic candidate for the prevention or treatment of tardive dyskinesia (TD) in clinical settings. However, further detailed research is required to confirm these preclinical findings and fully elucidate VTX’s therapeutic potential in human studies.

## 1. Introduction

Tardive dyskinesia (TD) is a hyperkinetic movement disorder induced by medical treatment, primarily affecting the orofacial region. It manifests as involuntary choreiform, athetoid, and rhythmic movements of the mouth, face, and tongue. TD typically emerges as a late-onset side effect following the chronic treatment of neuroleptic medications, especially first-generation antipsychotics, and poses a significant clinical challenge in the management of schizophrenia. This condition can persist for months or even years after the cessation of the causative drug, with approximately 2% of cases being irreversible [1,2,3].

Haloperidol (HPD), a first-generation butyrophenone neuroleptic, is commonly used to manage schizophrenia. Administration of HPD over 21 days leads to dopamine (DA) D2 receptor blockade, causing neurotoxicity and inducing orofacial dyskinesia (OD) in animal models. This condition is characterized by behaviors such as vacuous chewing movements (VCMs), tongue protrusion (TP), orofacial bursts, and cataleptic behavior, and has long served as a valuable model for studying the neuropathology of TD [4].

The OD induced by HPD in rats is strongly linked to an increase in striatal nitric oxide (NO) production, byproducts of lipid peroxidation (LPO), and inflammatory mediators, as well as a marked decrease in antioxidative enzyme activity. These factors collectively contribute to neurodegeneration and motor dysfunction, which resemble Parkinson’s disease (PD) symptoms, consistent with clinical observations [5,6,7,8,9,10,11,12,13]. Although the precise pathophysiological mechanisms underlying OD remain unclear, current research suggests that treatment with antioxidant or anti-inflammatory agents may provide therapeutic benefits for TD in animal models.

Vitexin (VTX), a C-glycosylated flavone (5, 7, 4-trihydroxyflavone-8-glucoside) and a significant bioactive component of the traditional Chinese herb *Crataegus pinnatifida* (hawthorn), has demonstrated a favorable safety profile in both animal and human studies [14]. Recent preclinical research has highlighted VTX’s broad therapeutic potential, including its antibacterial, antiviral, anti-inflammatory, antidiabetic, antitumor, and cardioprotective properties [14,15]. VTX, as a well-known strong antioxidant, can effectively scavenge reactive radicals, increase endogenous antioxidants, decrease peroxidative reactions, and preserve catalase activity within the mitochondria, probably by the nuclear factor erythroid-2-related factor 2 (Nrf2)-mediated pathway. These antioxidant properties contribute to its neuroprotective effects against various behavioral impairments, including seizures, depression-like symptoms, and motor and memory dysfunctions caused by ischemia, toxins, and stress [14,15,16,17,18,19,20,21,22]. Despite the substantial research on VTX’s benefits, its potential as a neuroprotectant against HPD-induced OD and the mechanisms behind this effect have yet to be explored.

Nrf2, a basic leucine zipper transcription factor, regulates the expression of antioxidant enzymes by binding to antioxidant response elements (AREs). Under normal conditions, Nrf2 is bound to its negative regulator, Kelch-like ECH-associated protein 1 (Keap1), which targets it for proteasomal degradation. However, during oxidative stress, reactive oxygen species (ROS) bind to Keap1’s cysteine residues, triggering a structural change that prevents Nrf2 from being ubiquitinated and degraded. This allows Nrf2 to translocate into the nucleus, where it activates the expression of antioxidant genes by binding to AREs. Elevated Nrf2 levels help counteract the effects of ROS, maintain redox homeostasis, and support cell survival. The Nrf2/Keap1 signaling pathway plays a critical role in regulating oxidative stress and is involved in key processes such as inflammation, endothelial dysfunction, cancer, and neurodegenerative diseases such as PD [23,24].

Based on VTX’s established antioxidative, anti-inflammatory, anti-apoptotic, and neuroprotective properties, it is hypothesized that VTX may reduce HPD-induced nitrative and oxidative damage, mitochondrial dysfunction, neuroinflammation, and apoptosis in the striatum, thereby preventing the onset of OD in experimental animal models [14,15,18,19,20,21,22]. This study aimed to assess the potential therapeutic effects of VTX and its involvement in the Nrf2-mediated pathway using a well-established rat model of HPD-induced neurotoxicity. The study concentrated on several critical aspects: (1) the increase in VCM and TP related to OD; (2) nitrative and oxidative stress, assessed by measuring NO and LPO byproducts; (3) antioxidant capacity, evaluated through the activities of superoxide dismutase (SOD), glutathione (GSH), and catalase (CAT); (4) mitochondrial function, determined by the activities of succinate dehydrogenase (SDH), ATPase, NADH–cytochrome C reductase, and succinate–cytochrome C reductase; (5) neuroinflammation, measured by the levels of interleukin-1β (IL-1β), tumor necrosis factor α (TNF-α), and interleukin-6 (IL-6); and (6) apoptosis, as estimated by caspase-3 activity in the striatum.

This study builds on existing research by linking HPD-induced OD with changes in striatal myristylation, oxidative stress, neuroinflammation, mitochondrial dysfunction, and neurodegeneration [5,6,7,8,9,10,11,12,13,19]. To further investigate the role of the Nrf2 pathway in this context, the study employed a combination of trigonelline (TGN), an inhibitor of this pathway, and VTX.

## 2. Results

Before the administration of HPD, the tested parameters showed no significant differences between the different groups. Tukey’s test confirmed that there were no significant variations between the V10 or V30 groups and the C group (V10 orV30 vs. C, *p* > 0.05).

### 2.1. Impact of VTX on Increases in VCM and TP Induced by HPD

HPD administration led to a substantial increase in VCM (Figure 1a) and TP (Figure 1b) frequency across days 1, 7, 14, and 21. Specifically, HPD administration caused a significant rise in VCM (H vs. C, *p* < 0.001) and TP (H vs. C, *p* < 0.001) counts on day 1, day 7, day 14, and day 21 as compared to the C group. VTX (10 mg/kg and 30 mg/kg) did not cause significant changes in VCM and TP frequencies on days 1 and 7 (V10 or V30 vs. H, *p* > 0.05). However, by day 21, 10 mg/kg VTX significantly reduced VCM and TP counts (VCM: decreased by 29.21%, V10 vs. H, *p* < 0.001; TP: decreased by 30.14%, V10 vs. H, *p* < 0.001). VTX at 30 mg/kg notably reduced VCM and TP counts on day 14 (VCM: decreased by 26.47%, V30 vs. H, *p* < 0.001; TP: decreased by 35.53%, V30 vs. H, *p* < 0.001) and day 21 (VCM: decreased by 46.35%, V30 vs. H, *p* < 0.001; TP: decreased by 45.48%, V30 vs. H, *p* < 0.001). The co-administration of TGN significantly mitigated the effects of VTX on HPD-induced VCM and TP changes, indicating that the therapeutic effect of VTX on HPD-induced orofacial movement disorders may be mediated through the Nrf2 pathway.

### 2.2. Impact of VTX on Increases in Striatal Nitric Oxide (NO) and Lipid Peroxide Production Induced by HPD

HPD treatment significantly increased striatal NO levels, measured as nitrites, from 112.86 ± 6.72 to 268.86 ± 9.89 μg/mL (H vs. C, *p* < 0.001) (Figure 2a) and elevated thiobarbituric acid reactive substance (TBARS) levels from 30.14 ± 2.91 to 62.14 ± 4.74 nmol/mg protein (H vs. C, *p* < 0.001) (Figure 2b) by day 21. The increases in both nitrite and TBARS levels in HPD-treated rats were reduced by VTX treatment significantly. At 10 mg/kg, VTX decreased nitrite levels by 44.69% (V10 vs. H, *p* < 0.001) and TBARS levels by 45.53% (V10 vs. H, *p* < 0.001). At 30 mg/kg, VTX reduced nitrites by 63.28% (V30 vs. H, *p* < 0.001) and TBARS levels by 72.75% (V30 vs. H, *p* < 0.001). The co-administration of TGN nearly nullified VTX’s protective effects against the HPD-induced increases in striatal NO and lipid peroxide levels.

### 2.3. Impact of VTX on Reductions in Striatal Antioxidant Power Induced by HPD

HPD administration led to significant reductions in striatal levels of GSH (Figure 3a), SOD (Figure 3b), and CAT (Figure 3c) by day 21. Specifically, GSH decreased from 14.63 ± 0.76 to 5.94 ± 0.73 nmol/mg tissue (H vs. C, *p* < 0.001), SOD from 2.87 ± 0.15 to 1.54 ± 0.13 U/mg tissue (H vs. C, *p* < 0.001), and CAT from 8.26 ± 0.71 to 2.14 ± 0.43 U/mg tissue (H vs. C, *p* < 0.001). VTX treatment at 10 mg/kg significantly mitigated these reductions, with GSH levels increasing by 41.43% (V10 vs. H, *p* < 0.001), SOD rising by 48.12% (V10 vs. H, *p* < 0.001), and CAT increasing by 45.1% (V10 vs. H, *p* < 0.001). At 30 mg/kg, VTX showed even greater improvements: GSH increased by 73.53% (V30 vs. H, *p* < 0.001), SOD by 69.92% (V30 vs. H, *p* < 0.001), and CAT by 73.04% (V30 vs. H, *p* < 0.001). The protective effects of VTX were significantly reduced when TGN was co-administered, indicating that VTX’s benefits against HPD-induced declines in striatal GSH, SOD, and CAT levels are mediated through the Nrf2 pathway.

### 2.4. Impact of VTX on Striatal Mitochondrial Dysfunction Induced by HPD

By day 21, HPD treatment had significantly reduced key mitochondrial enzymes and markers in the striatum. Specifically, the optical density (OD) (H vs. C, *p* < 0.001), total ATPase activity (Figure 4b) (H vs. C, *p* < 0.001), NADH–cytochrome C reductase activity (Figure 4c) (H vs. C, *p* < 0.001), and succinate–cytochrome C reductase activity (Figure 4d) (H vs. C, *p* < 0.001).

VTX treatment at 10 mg/kg and 30 mg/kg significantly reversed the declines observed. At 10 mg/kg, VTX restored SDH levels (490 nm/mg protein) (V10 vs. H, *p* < 0.001), total ATPase activity (V10 vs. H, *p* < 0.01), NADH–cytochrome C reductase (V10 vs. H, *p* < 0.001), and succinate–cytochrome C reductase (V10 vs. H, *p* < 0.001). At 30 mg/kg, these measures improved further, with SDH (V30 vs. H, *p* < 0.001), total ATPase (V30 vs. H, *p* < 0.001), NADH–cytochrome C reductase (V30 vs. H, *p* < 0.001), and succinate–cytochrome C reductase (V30 vs. H, *p* < 0.001). The co-administration with TGN significantly reduced VTX’s benefits, highlighting the Nrf2 pathway’s critical role.

### 2.5. Impact of VTX on Increases in Striatal Neuroinflammatory and Apoptotic Markers Induced by HPD

Treatment with HPD for 21 days led to significant increases in striatal levels of pro-inflammatory cytokines and apoptotic markers. Specifically, levels of TNF-α (Figure 5a) rose from 39.14 ± 4.06 to 116.29 ± 8.67 pg/mL protein (*p* < 0.001), IL-1β (Figure 5b) increased from 35.29 ± 3.35 to 106.29 ± 7.71 pg/mL protein (*p* < 0.001), IL-6 (Figure 5c) went up from 40.29 ± 3.04 to 112.29 ± 11.31 pg/mL protein (*p* < 0.001), and caspase-3 (Figure 5d), an apoptotic marker, increased from 1.81 ± 0.35 to 4.91 ± 0.34 nmol/mg protein (*p* < 0.001).

VTX treatment at 10 mg/kg significantly reduced these HPD-induced elevations. TNF-α levels decreased to 78.57 ± 6.45 pg/mL protein (*p* < 0.001), IL-1β to 70.86 ± 7.29 pg/mL protein (*p* < 0.001), IL-6 to 78.57 ± 6.02 pg/mL protein (*p* < 0.001), and caspase-3 to 3.53 ± 0.33 nmol/mg protein (*p* < 0.001). The 30 mg/kg dose of VTX further reduced these markers. TNF-α levels decreased to 63.29 ± 5.19 pg/mL protein (*p* < 0.001), IL-1β to 59.43 ± 7.41 pg/mL protein (*p* < 0.001), IL-6 to 65.71 ± 7.3 pg/mL protein (*p* < 0.001), and caspase-3 to 2.63 ± 0.34 nmol/mg protein (*p* < 0.001).

However, the co-administration of TGN significantly negated the protective effects of VTX on these neuroinflammatory and apoptotic markers, highlighting the critical role of the Nrf2 pathway in mediating the anti-inflammatory and anti-apoptotic effects of VTX against HPD-induced neuroinflammation and cell death.

## 3. Discussion

In this study, we demonstrated that VTX offers protective effects against haloperidol HPD-induced OD, neuroinflammation, nitrosative and oxidative damage, mitochondrial dysfunction, and apoptotic activation, potentially mediated through the Nrf2 pathway in experimental animals. To the best of our knowledge, this is the first report showing that VTX provides robust protection against HPD-induced pathophysiological dysfunctions, highlighting its potential therapeutic value for future clinical studies.

VTX, a flavonoid with notable neuroprotective properties, can be found in various dietary sources. Common sources include hawthorn (*Crataegus pinnatifida*), which is a significant source of VTX often used in traditional medicine. Additionally, eggplant (Solanum melongena), particularly its skin, contains VTX. Strawberries (Fragaria × ananassa), certain varieties of which also have vitexin, along with red quinoa (*Amaranthus cruentus*), which provides vitexin in its seeds and leaves, contribute to dietary intake. Mint (*Mentha* spp.) and black goji berries (*Lycium ruthenicum*) are also known to contain VTX. Incorporating these foods into the diet can help increase VTX consumption, potentially benefiting overall health [14,15,21,22].

The alterations in VCM and TP induced by HPD in rats serve as a well-established model for investigating potential therapeutic agents for TD [4,5,6,7,10,11,12,13]. The characteristic features of HPD-induced OD were closely similar to the symptoms of TD, and HPD has been linked to TD development in humans, despite ongoing debates about the validity of animal models for TD. Our findings are consistent with previous studies, demonstrating that 21 consecutive days of HPD treatment significantly induced OD in rats, as evidenced by an increased frequency of VCM and TP, together with the impairments in oxidative defense and mitochondrial function. These changes, resulting from striatal degeneration, underscore the neurotoxic effects of HPD [5,6,10,11,12,13].

HPD-induced OD is closely linked to striatal nitrosative and oxidative stress, as well as inflammatory responses [4,5,6,10,11,12,13]. In this study, we found that HPD treatment led to increased levels of nitrite and TBARSs, a reduction in GSH (an endogenous antioxidant), and decreased activities of the cellular antioxidant enzymes SOD and CAT. Additionally, HPD negatively affected the levels of SDH, ATPase, and electron transport chain (ETC) enzymes in the rat striatum. These results underscore the contribution of nitric oxide (NO) and free radical-induced toxicity, along with mitochondrial dysfunction, in the development of HPD-induced OD.

HPD, a first-generation butyrophenone neuroleptic, acts as a dopamine (DA) antagonist by blocking D2 receptors, which leads to an increased DA turnover. This elevated turnover results in the production of reactive metabolites and hydrogen peroxide, thereby raising oxidative stress in dopaminergic neurons [6,7,12,13,25,26]. Additionally, the autoxidation of DA to o-quinone aminochrome, which can further oxidize to leukoaminochrome o-semiquinone radicals, is a significant source of endogenous reactive species. HPD-induced DA turnover, combined with increased glutamatergic transmission, exacerbates free radical generation and oxidative stress [25,26]. Due to the high concentration of monoamines in the striatum, it is particularly vulnerable to free-radical damage and oxidative stress.

Inhibition of the electron transport chain in the inner mitochondrial membrane by HPD interrupts mitochondrial respiration, leading to an increase in reactive oxygen species (ROS) [12,26,27,28]. This mitochondrial dysfunction, coupled with ATP depletion and reduced Na^+^/K^(+)^-ATPase activity, results in neuronal depolarization, which alleviates the voltage-dependent Mg^2+^ block of NMDA receptors, further impairing mitochondrial function [29]. The subsequent hyperexcitability, mediated by NMDA receptor activation, causes a substantial influx of calcium ions (Ca^2+^), along with the production of ROS and reactive nitrogen species (RNS), leading to lipid peroxidation and injury to both mitochondrial and nuclear DNA [29,30]. The interplay between mitochondrial dysfunction and NMDA receptor activation exacerbates the nitrative and oxidative stress induced by HPD. Additionally, nitric oxide (NO) has been shown to inhibit key enzymes of energy metabolism, exacerbating damage in conditions characterized by heightened ROS and RNS production [11,12,13,31]. These findings collectively highlight the crucial roles of nitrative and oxidative stress, along with mitochondrial dysfunction, in the development of HPD-induced OD.

Immoderate nitrative and oxidative stress can initiate an inflammatory reaction, leading to the inflammatory mediators’ increase. These mediators activate apoptotic pathways, which are crucial for neuronal cell death and play a significant role in the development of OD [5,10,12,13]. In our study, TNF-α, IL-1β, and IL-6 increased in the rat striatum of the H group, consistent with previous findings. This suggests that the neuroinflammatory response triggered by HPD-induced neurotoxicity contributes to striatal damage and OD development. Additionally, increased levels of caspase-3, an important enzyme in the apoptotic process, were observed following HPD treatment, reinforcing the role of apoptosis in the pathology of OD in the rat striatum. These observations are in line with preclinical studies indicating that prolonged HPD exposure leads to changed neuronal activity in the striatum because of neuronal injury or death [5,8,9,12,13,32].

In clinical practice, TD is commonly treated with a combination of antipsychotics and anticholinergic drugs like biperiden. However, these treatments can lead to anticholinergic side effects such as tachycardia, mydriasis, dry mucous membranes, and urinary retention, and may also worsen positive symptoms of schizophrenia [1,2,3]. This underscores the need for alternative therapeutic approaches for TD. VTX, a key bioactive compound from the traditional Chinese herb [*Crataegus pinnatifida* (hawthorn)], has been demonstrated to possess strong antioxidative, anti-inflammatory, and antiapoptotic properties. In addition, it has been shown to prevent neurochemical deficits, offering neuroprotection and serving as a safeguard against HPD-induced OD [14,15,18,19,20,21,22].

Our results show that VTX successfully decreased the elevated levels of nitrite, TBARSs, TNF-α, IL-1β, IL-6, and caspase-3, and at the same time, increased levels of SDH, ATPase, ETC enzymes, GSH, SOD, and CAT in the rat striatum after HPD treatment. However, TGN significantly diminished VTX’s protective effects, suggesting that VTX’s neuroprotective benefits against HPD-induced OD are likely due to its ability to neutralize excess NO and free radicals, improve antioxidant defenses, sustain mitochondrial function, and inhibit inflammation and apoptotic pathways in the striatum, potentially through interactions with the Nrf2-mediated pathway.

The antioxidant properties of VTX are attributed to its ability to directly scavenge oxygen free radicals and protect antioxidant enzymes [14,15,18,20,21]. VTX has been shown to donate electrons via the adjacent dihydroxyl structure of the A ring, making it an effective radical scavenger. Additionally, VTX upregulates the expression of Nrf2 and enhances the activities of GSH and antioxidant enzymes such as SOD, CAT, GPx, and GST. VTX is also believed to inhibit the NF-kB transcription factor, which downregulates pro-inflammatory mediators like TNF-α and IL-1β while upregulating anti-inflammatory mediators such as IL-4 and IL-10, thus exerting anti-inflammatory effects [14,20,21,22].

Furthermore, VTX activates the PI3K/Akt, ERK1/2, and protein kinase C (PKC) pro-survival signaling pathways, which upregulate the release of anti-apoptotic proteins such as Bcl2, BDNF, and Nrf2 while downregulating the expression of pro-apoptotic proteins like caspase-3, caspase-9, Bax, and Bad, among others [18,19,21]. This activation contributes to neuroprotection. Based on these findings, we strongly believe that VTX is a promising candidate for protection against HPD-induced neuronal oxidation, inflammation, apoptosis, and behavioral impairments.

Nrf2, a transcription factor that regulates genes involved in antioxidation, inflammation, and cell survival, has a pivotal role in controlling oxidative and inflammatory responses in the experimental models of PD [21,33,34]. Nrf2 facilitates the increase in antioxidative proteins and suppresses the expression of inflammatory cytokine genes, thereby lowering reactive oxygen species (ROS), inflammation, and apoptosis. It is also essential for preserving mitochondrial function in in vitro PD models [21,33,34,35,36]. Our findings support the idea that vitexin (VTX) exerts its protective effects through the Nrf2 pathway. However, the inhibition of Nrf2 negates these protective benefits [18,19,21]. These effects are linked to the induction of heme oxygenase-1 (HO-1) via Nrf2 [18]. Additionally, the PI3K/Akt and p38 pathways, which act upstream, play roles in activating Nrf2. The involvement of the PI3K/Akt pathway in Nrf2 activation and its contribution to the neuroprotective effects of VTX has been observed [19]. Further research is needed to fully understand how VTX modulates the Nrf2 pathway.

## 4. Materials and Methods

### 4.1. Experimental Animals

All experiments followed the NIH Guide for the Care and Use of Laboratory Animals and were approved by the IACUC at National Taiwan University School of Medicine (approval number: 20200513). Wistar rats (270–300 g, ~3 months old) from BioLASCO Taiwan Co., Ltd. (Taipei, Taiwan) were housed in groups of three with free access to food and water in a 22 ± 3 °C environment with a 12 h light/dark cycle (lights on at 7:00 AM). To reduce anxiety, rats were handled gently for 20 min daily for four days before the experiments.

### 4.2. Drugs

Haloperidol (HPD, CAS: 52-86-8, H1512-10G), vitexin (VTX, ≥95%; CAS: 3681-93-4), and trigonelline (TGN, CAS: 535-83-1, 1686411) were sourced from Sigma (St. Louis, MO, USA). The drugs were dissolved in normal saline and administered intraperitoneally (i.p.) daily for 21 days with freshly prepared solutions. Dosages were based on prior studies [6,11,12,13,18,20,37] and administered at 2.0 mL/kg body weight. VTX was initially tested at 1 mg/kg with no significant effects; the dosage was incrementally increased to a maximum of 100 mg/kg to achieve statistically significant results, but significant results were observed at 10, 30, and 100 mg/kg, particularly in reducing HPD-induced OD. However, there were no significant variations at 30 and 100 mg/kg. Therefore, 10 and 30 mg/kg were used in this study.

### 4.3. Experimental Design and Procedure

Rats were randomly divided into eight groups of eight rats per treatment group (n = 8), with an equal representation of both sexes, as outlined in Table 1. To induce OD, HPD was administered at a dose of 1 mg/kg intraperitoneally (i.p.) for 21 consecutive days. Depending on the group, VTX at doses of 10 or 30 mg/kg was administered i.p. 60 min after the HPD injection, also for 21 consecutive days. In groups where TGN was used, it was injected 30 min before VTX administration. The experimental paradigm is depicted in Figure 6.

### 4.4. Behavioral Assessment of Orofacial Dyskinesia (OD)

The behavior associated with OD in all rats was evaluated and quantified according to protocols established in prior studies [12,13] and standard procedures routinely used in our laboratory. These assessments were conducted 6 h after the administration of HPD or normal saline on days 1, 7, 14, and 21. To minimize bias, each rat was randomly assigned a number, and two experienced researchers who were blinded to the treatment groups independently performed the behavioral assessments. The rats were placed individually in an assessment cage (20 cm × 20 cm × 19 cm) that included mirrors beneath the floor, enabling the observation of behavior even when the rat was not facing the observer. The frequency of OD, vacuous chewing movements (VCMs), and tongue protrusions (TP) was recorded over a 5 min period following a 2 min adaptation period. All behavioral experiments took place between 09:00 a.m. and 11:00 a.m.

### 4.5. Biochemical Analysis

On day 21, one hour after the final behavioral assessment, the rats were euthanized. The brains were immediately extracted, rinsed with ice-cold saline to remove any residual blood, and stored at −80 °C. The striatum was quickly dissected on an ice-cold surface, using the stereotaxic atlas from Budantsev et al. [38] as a guide. The isolated striatal tissue was washed with isotonic saline, weighed, and homogenized in 0.1 N HCl. A 10% (*w*/*v*) tissue homogenate was then prepared using a 0.1 M phosphate buffer (pH 7.4). For the catalase (CAT) assay, the postnuclear fraction was obtained by centrifuging the homogenate at 1000× *g* for 20 min at 4 °C. For the assays of the remaining enzymes, the homogenates were centrifuged at 12,000× *g* for 60 min at 4 °C.

#### 4.5.1. Estimation of Nitrite 

Nitrite concentrations, the end products of nitric oxide (NO) metabolism, were measured using Roche’s “NO colorimetric assay” [39] (23,479, Sigma, St. Louis, MO, USA). This assay involves reacting nitrite with sulfanilamide and N-(1-naphthyl)-ethylenediamine dihydrochloride to produce a reddish-violet dye, quantified spectrophotometrically at 540 nm. The procedure included mixing 100 μL of homogenate with 400 μL re-distilled water, heating at 100 °C for 15 min, cooling, and adding Carrez I and II reagents. The mixture was alkalized to pH 8.0, centrifuged, and 75 μL of the supernatant was combined with 75 μL re-distilled water in a microplate. After incubation at 25 °C for 30 min, absorbance was measured. The reaction was then developed with sulfanilamide and N-(1-naphthyl)-ethylenediamine dihydrochloride, and absorbance was measured again. Nitrite levels were calculated from a standard curve of sodium nitrite concentrations (6–600 μM) and expressed as μg/mL.

#### 4.5.2. Estimation of Lipid Peroxidation

Lipid peroxide concentration was assessed using the thiobarbituric acid reactive substance (TBARS) assay (MAK085, Sigma, St. Louis, MO, USA), as described by Ohkawa et al. [40]. This method measures the concentration of malondialdehyde (MDA), a marker of lipid peroxidation. Tissue homogenates were reacted with thiobarbituric acid and the resultant color was measured spectrophotometrically. The MDA concentration, indicative of lipid peroxidation, was expressed in nanomoles of malondialdehyde per milligram of protein (nmol MDA/mg protein). In addition, MDA was normalized to a standard preparation of 1,1,3,3-tetra ethoxypropane.

#### 4.5.3. Estimation of Reduced Glutathione (GSH)

GSH (MAK364, Sigma, St. Louis, MO, USA) levels were measured according to Ellman’s method [41]. After precipitating proteins with 10% trichloroacetic acid and centrifuging, the supernatant was mixed with Ellman’s reagent. The resulting yellow color was measured at 412 nm, with GSH concentration expressed as nmol GSH/mg tissue.

#### 4.5.4. Estimation of Superoxide Dismutase (SOD) 

SOD (19,161, Sigma, St. Louis, MO, USA) activity was measured by its ability to inhibit the oxidation of epinephrine to adrenochrome, following Misra and Fridovich’s method [42]. The reaction mixture included 0.05 mL tissue supernatant, 2.0 mL carbonate buffer, and 0.5 mL ethylenediaminetetraacetic acid (EDTA), with the reaction initiated by 0.5 mL epinephrine. Autooxidation at pH 10.2 was monitored by optical density at 480 nm every minute. Results were expressed as nmol SOD/mg tissue, with one unit defined as the amount needed to inhibit 50% of epinephrine oxidation.

#### 4.5.5. Estimation of Catalase (CAT) 

CAT (CAT100, Sigma, St. Louis, MO, USA) activity was measured using Beers and Sizer’s method [43]. The reaction mixture included 2 mL phosphate buffer, 0.95 mL hydrogen peroxide (0.019 M), and 0.05 mL tissue supernatant. Hydrogen peroxide decomposition was monitored by absorbance at 240 nm every 10 s for 1 min. One unit of CAT activity is defined as the amount of enzyme needed to decompose 1 mmol of hydrogen peroxide per minute at 25 °C and pH 7.0. Results are given as units of CAT activity per milligram of protein (units/mg tissue).

### 4.6. Estimation of Mitochondrial Function

Mitochondrial function was evaluated using differential centrifugation [44]. A 10% striatal homogenate in 0.25 M ice-cold tris–sucrose buffer was first centrifuged at 1000× *g* for 10 min to isolate the nuclear pellet. The supernatant was then subjected to centrifugation at 10,000× *g* for 20 min to collect the mitochondrial pellet, which was washed three times with mannitol–sucrose–HEPES buffer before resuspension.

#### 4.6.1. Succinate Dehydrogenase (SDH) 

SDH (MAK197, Sigma, St. Louis, MO, USA) activity was assessed using a modified Pennington method [45]. A 0.05 mg of mitochondrial protein was incubated for 10 min with 50 mM potassium phosphate, 0.01 M sodium succinate, and 2.5 µg/mL p-iodonitrotetrazolium violet. The reaction was halted with 10% trichloroacetic acid and the color was extracted using a mixture of ethyl acetate, ethanol, and TCA (5:5:1). SDH activity was determined by measuring the optical density at 490 nm per milligram of protein.

#### 4.6.2. Total ATPase 

Total ATPase (MAK113, Sigma, St. Louis, MO, USA) activity was evaluated by measuring inorganic phosphate released from ATP, following Prasad and Muralidhara’s method [46]. Cytosolic protein (50 µg) was incubated with 0.02 M tris–HCl buffer, including 100 mM NaCl, 20 mM KCl, and 5 mM MgCl_2_, at 37 °C for 15 min. The reaction was stopped with 20% TCA and the mixture was centrifuged at 15,000× *g* for 10 min. The phosphate concentration in the supernatant was measured, with an enzyme-free control as a blank. ATPase activity was expressed as micrograms of inorganic phosphate released per milligram of protein.

#### 4.6.3. NADH–Cytochrome C Reductase and Succinate–Cytochrome C Reductase

The activities of NADH–cytochrome C reductase (complex I-III) and succinate–cytochrome C reductase (complex II–III) (MAK360, Sigma, St. Louis, MO, USA) were determined following standard methods as outlined by Navarro et al. [47].

### 4.7. Estimation of Tumor Necrosis Factor-α (TNF-α), Interleukin-6 (IL-6), and Interleukin-1β (IL-1β)

The levels of TNF-α (KB2145), IL-6 (KB2068), and IL-1β (KB2063) were measured using ELISA kits from KRISHGEN BioSystem (Whittier, CA). The ELISA method used is a solid-phase sandwich assay that runs over 4.5 h to measure these cytokines in rat samples. Absorbance readings were taken with a microtiter plate reader and the concentrations were determined using standard curves. Results were reported in pg/mL of protein.

### 4.8. Estimation of Caspase-3

Caspase-3 activity, a marker of apoptosis, was assessed using the R&D Systems caspase-3 colorimetric kit (GTX85558) (GeneTex Inc., Hsinchu, Taiwan). This assay involved adding a peptide specific for caspase-3, conjugated to the color reporter molecule p-nitroaniline (pNA), to tissue lysates or homogenates. Caspase-3 cleaves the peptide, releasing pNA, which was then quantified by measuring absorbance at 405 nm. The results were reported in nmol of pNA per milligram of protein.

### 4.9. Estimation of Protein

The protein content in cytosolic and mitochondrial fractions was determined using the Lowry method [48], with bovine serum albumin (A3294, Sigma, St. Louis, MO, USA) as a standard.

### 4.10. Statistical Analysis

Results are presented as mean ± SEM. Statistical analyses were performed with GraphPad Prism 8.3.0. Behavioral data used repeated-measures two-way ANOVA and biochemical data used one-way ANOVA. Tukey’s test was used for post hoc analysis. A *p*-value < 0.05 was considered significant.

## 5. Conclusions

In summary, findings in this study provide strong evidence for the potential of VTX in treating OD in experimental animal models. The data suggest that VTX may offer neuroprotection through mechanisms such as reducing oxidative stress, preventing mitochondrial dysfunction, decreasing neuroinflammation, and inhibiting apoptosis, likely involving interactions with the Nrf2 pathway, which is crucial for cellular defense. Using HPD-induced OD in rats, which mirrors the features of TD in humans, establishes a solid foundation for exploring VTX as a treatment. Our study highlights VTX as a promising adjunct therapy, potentially addressing the safety and treatment challenges of TD. However, further cellular and molecular studies will be required to confirm the possible neuroprotective mechanisms of VTX and further research is needed to translate these preclinical results into clinical practice. Future clinical trials will be essential to confirm VTX’s effectiveness and safety in human populations, which could lead to improved treatment options for individuals suffering from this debilitating condition.

## Figures and Tables

**Figure 1 ijms-25-10206-f001:**
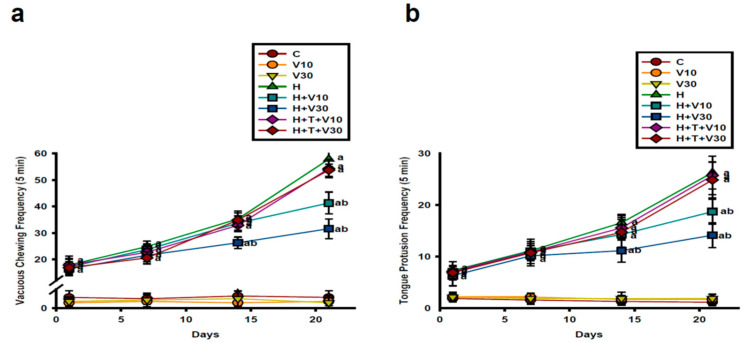
Impact of VTX on behaviors associated with HPD-induced orofacial dyskinesia (OD). (**a**) Vacuous chewing movements (VCMs) and (**b**) tongue protrusions (TP) observed in rats over days 1 to 21. Data are shown as mean ± SEM (*n* = 8). Data were analyzed by using two-way ANOVA with Tukey’s post hoc test: a corresponds to *p* < 0.001 vs. C; b corresponds to *p* < 0.001 vs. H.

**Figure 2 ijms-25-10206-f002:**
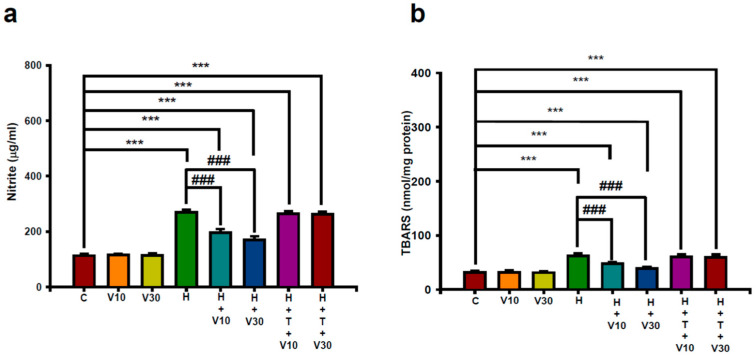
Impact of VTX on nitrosative and oxidative stress in the striatum induced by HPD was assessed by measuring (**a**) nitrite levels and (**b**) thiobarbituric acid reactive substances (TBARSs) in rats. Data are presented as mean ± SEM (*n* = 8). Data were analyzed by using one-way ANOVA with Tukey’s post hoc test to determine differences between groups. *** *p* < 0.001 vs. C; ### *p* < 0.001 vs. H.

**Figure 3 ijms-25-10206-f003:**
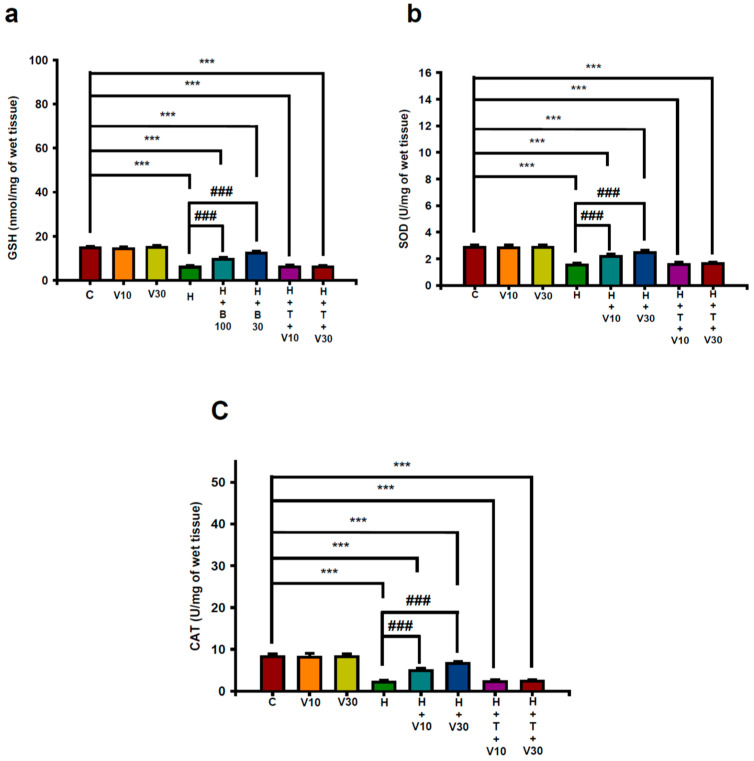
Impact of VTX on the reduction in striatal antioxidative capacity induced by HPD. (**a**) Glutathione (GSH), (**b**) superoxide dismutase (SOD), and (**c**) catalase (CAT) in rats. Data are shown as mean ± SEM (*n* = 8). Statistical analysis was performed using one-way ANOVA with Tukey’s post hoc test: *** *p* < 0.001 vs. C; ### *p* < 0.001 vs. H.

**Figure 4 ijms-25-10206-f004:**
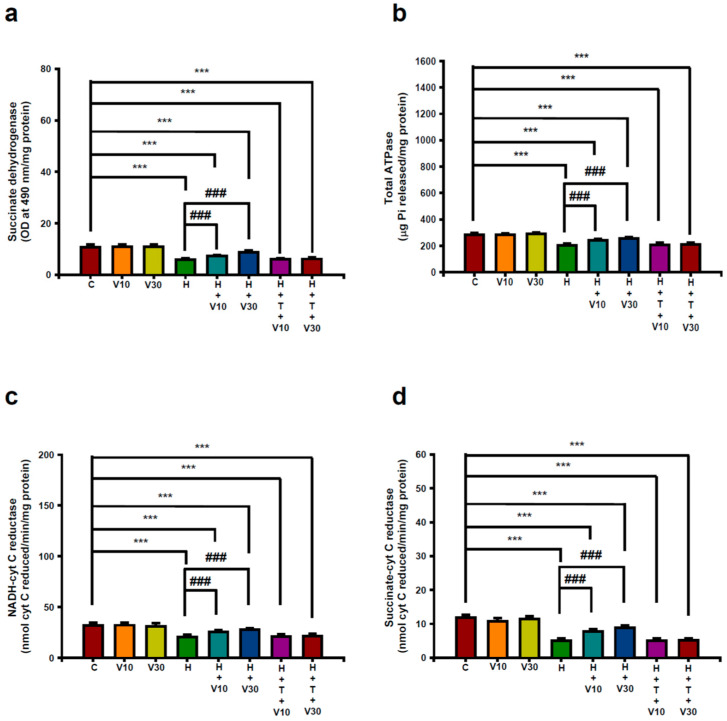
The impact of VTX on HPD-induced mitochondrial dysfunction in the striatum, covering (**a**) succinate dehydrogenase (SDH), (**b**) total ATPase, (**c**) NADH–cytochrome c reductase (complexes I–III), and (**d**) succinate–cytochrome c reductase (complexes II–III). The data are expressed as mean ± SEM (*n* = 8). Data were analyzed by using one-way ANOVA followed by Tukey’s post hoc test: *** *p* < 0.001 vs. C; ### *p* < 0.001 vs. H.

**Figure 5 ijms-25-10206-f005:**
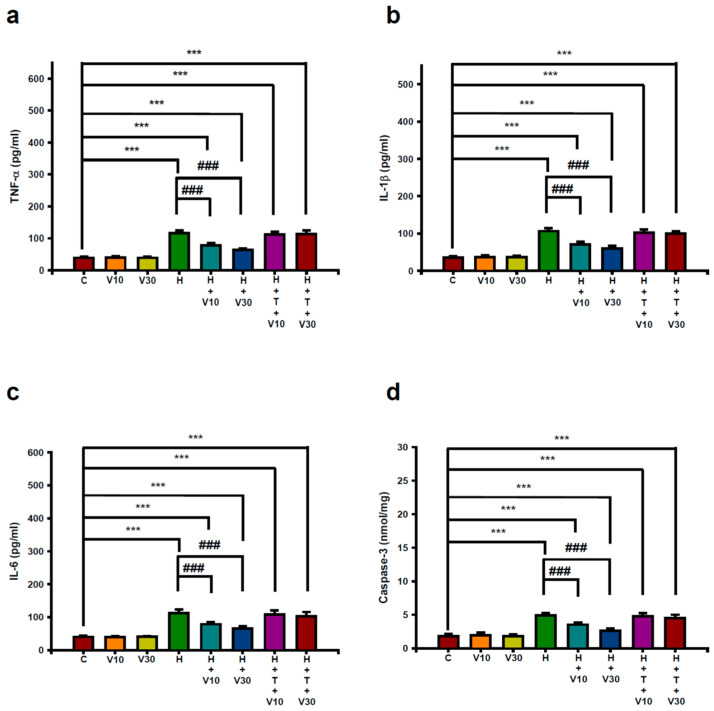
The VTX effects on HPD-induced increases in striatal neuroinflammation and apoptotic markers. (**a**) TNF-α, (**b**) IL-1β, (**c**) IL-6, and (**d**) caspase-3. The data are expressed as mean ± SEM (*n* = 8). Post hoc Tukey’s test after one-way ANOVA. *** *p* < 0.001 vs. C; ### *p* < 0.001 vs. H.

**Figure 6 ijms-25-10206-f006:**
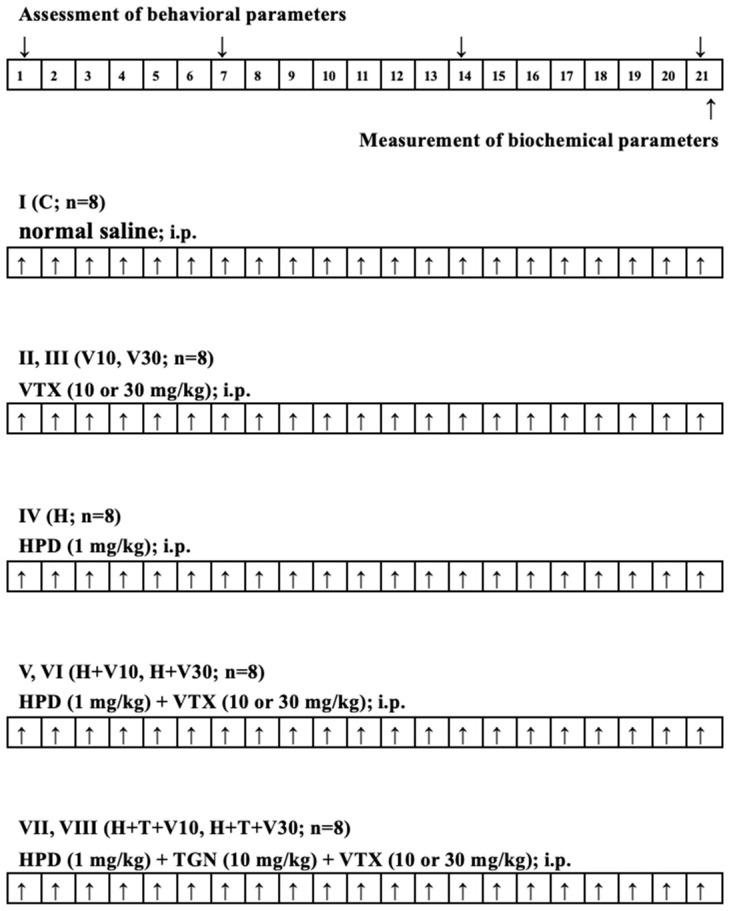
Experimental design and drug treatment paradigm.

**Table 1 ijms-25-10206-t001:** Experimental Groups.

Experimental Groups		
Experimental	Treatment	n = 8
Group		♂:♀ = 4:4
I.	C	control (normal saline; i.p.) for 21 days
II.	V10	VTX (10 mg/kg; i.p.) for 21 days
III.	V30	VTX (30 mg/kg; i.p.) for 21 days
IV.	H	HPD (1 mg/kg; i.p.) for 21 days
V.	H + V10	HPD (1 mg/kg; i.p.) + VTX (10 mg/kg; i.p.) for 21 days
VI.	H + V30	HPD (1 mg/kg; i.p.) + VTX (30 mg/kg; i.p.) for 21 days
VII.	H + T + V10	HPD (1 mg/kg; i.p.) + TGN (10 mg/kg; i.p.) + VTX (10 mg/kg; i.p.) for 21 days
VIII.	H + T + V30	HPD (1 mg/kg; i.p.) + TGN (30 mg/kg; i.p.) + VTX (10 mg/kg; i.p.) for 21 days

Note: HPD: haloperidol; i.p.: intraperitoneally; TGN: trigonelline; VTX: vitexin.

## Data Availability

The data presented in this study are available upon request from the corresponding author. The data are not publicly available due to privacy considerations.

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
