# Peer review of "Vitexin Mitigates Haloperidol-Induced Orofacial Dyskinesia in Rats through Activation of the Nrf2 Pathway"

_ijms, 2024, doi:10.3390/ijms251810206_

Round 1
Reviewer 1 Report
Comments and Suggestions for Authors
In this manuscript authors investigated the protective effects of VTX against orofacial dyskinesia in rats and found that VTX significantly reduced HPD-induced OD, decreased oxidative stress, enhanced antioxidant capacity, prevented mitochondrial dysfunction, and reduced neuroinflammatory and apoptotic markers in the striatum. Moreover, the protective effects of VTX on both behavioral and biochemical aspects of HPD-induced OD were significantly reduced when trigonelline was administered.
The manuscript is interesting and generally well written. However, there are several points that need an improvement. In particular:
Introduction: Although NRF2 plays an important role in this study, this transcription factor is just mentioned in the introduction. However, the multifaceted role of this transcription factor deserves to be highlighted since it plays a key role in the onset and progression of several diseases (see PMID: 37296999, PMID: 39034715). This is an important point to add since it can further highlight th einteresting results obtained by the authors.
I suggesst to replace letters with asterisks in the figures to show the statistical difference. Moreover, bars reporting the groups compared would be helpful to better understand the results obtained.
Figure 5: Y axis labeling should be reported just as "ng/mL"and "nmol/mg". Moreover, cleaved caspase 3 (the active form of caspase 3) should be assessed by western blot.
Line 278: GSH is not an enzyme
Table 1 must be formatted according to the journal style
4.2 Drugs: Authors must add the product code of all reagents and kits used to ensure data reproducibility.
Abbreviations must be written in full length when mentioned for the first time
Author Response
Response to Reviewer 1 Comments
Dear Reviewers,
I’m writing in response to your feedback regarding the manuscript we submitted. Thank you so much for your positive comments and suggestions for the manuscript entitled. The manuscript has been revised based on your inquiries, rephrased the content of the manuscript and resubmitted through the journal website. The revised parts have been marked in red. The following is the response to your inquiries point-by-point; we hope our responses fully address your comments and suggestions:
Reviewers’ comments:
In this manuscript authors investigated the protective effects of VTX against orofacial dyskinesia in rats and found that VTX significantly reduced HPD-induced OD, decreased oxidative stress, enhanced antioxidant capacity, prevented mitochondrial dysfunction, and reduced neuroinflammatory and apoptotic markers in the striatum. Moreover, the protective effects of VTX on both behavioral and biochemical aspects of HPD-induced OD were significantly reduced when trigonelline was administered.
Response:
Thank you so much for the kind comment and wonderful suggestion.
The manuscript is interesting and generally well written. However, there are several points that need an improvement. In particular:
Point 1:
Introduction: Although NRF2 plays an important role in this study, this transcription factor is just mentioned in the introduction. However, the multifaceted role of this transcription factor deserves to be highlighted since it plays a key role in the onset and progression of several diseases (see PMID: 37296999, PMID: 39034715). This is an important point to add since it can further highlight the interesting results obtained by the authors.
Response 1:
Thank you for your suggestions. We have made the necessary revisions based on your recommendations, added background information about Nrf2 and cited them in the revised manuscript. As you mentioned, we believe this addition enhances the readability and appeal of the article.
Point 2:
I suggest to replace letters with asterisks in the figures to show the statistical difference. Moreover, bars reporting the groups compared would be helpful to better understand the results obtained.
Response 2:
Thank you so much for giving us such a detail constructive comments and suggestions, we have replaced letters with asterisks in the figures to show the statistical difference and bars reporting the groups to be helpful to better understand the results obtained.
Point 3:
Figure 5: Y axis labeling should be reported just as "ng/mL"and "nmol/mg". Moreover, cleaved caspase 3 (the active form of caspase 3) should be assessed by western blot.
Response 3:
Thank you so much for suggestions again, we have corrected these in the revised manuscript.
Point 4:
Line 278: GSH is not an enzym
Response 4:
Thank you so much, we have corrected it in the revised manuscript.
Point 5:
Table 1 must be formatted according to the journal style
Response 5: Thank you so much for giving us such detail constructive comments and suggestions, we have changed Table 1 to be formatted according to the journal style in the revised manuscript.
Point 6:
4.2 Drugs: Authors must add the product code of all reagents and kits used to ensure data reproducibility.
Response 6: Thank you again for the constructive suggestions, we have added the product code of all reagents and kits used to ensure data reproducibility.
Point 7:
Abbreviations must be written in full length when mentioned for the first time
Response 7:
Thank you so much for giving us such detail constructive comments and suggestions, abbreviations have been written in full length when mentioned for the first time in the revised manuscript.
Thank you for your valuable comments/suggestions and giving us the opportunity to revise the manuscript to a more readable level. We worked very hard to response your inquiries and to revise the manuscript. Before we finalized the revised manuscript, we have resent the manuscript for proof-reading as well as one last review by a native English-speaking researcher.We hope the manuscript could pass the review to be published in your prestigious journal: International Journal of Molecular Sciences
Sincerely yours,
Hsiang-Chien Tseng
Department of Anesthesiology, Shin Kong Wu Ho-Su Memorial Hospital,
No. 95, Wen Chang Road, Shih Lin District, Taipei 11101, Taiwan.
Fax: +886 2 833 2211.
E-mail address: [email protected] (Hsiang-Chien Tseng).
Reviewer 2 Report
Comments and Suggestions for Authors
Manuscript ID- ijms-3216895
The paper entitled “Vitexin Mitigates Haloperidol-Induced Orofacial Dyskinesia in Rats Through Activation of the Nrf2 Pathway” has investigated the protective effects of Vitexin (VTX), a C-glycosylated flavone against orofacial dyskinesia (OD) in rats, induced by haloperidol (HPD), along with the neuroprotective mechanisms underlying these effects. Authors have performed the work with utmost scientific rigor and the work is commendable.
Comments-
In Figure 1- colored symbol for the data points may be useful for better visualization.
In the discussion of results- the following details reproduced in the materials and methods at line 318- 320 may be included- “VTX was initially tested at 1 mg/kg with no sig-318 nificant effects, but significant results were observed at 10 and 30 mg/kg, particularly in 319 reducing HPD-induced OD. Therefore, 10 and 30 mg/kg were used in this study.” Based on the above results what is the optimal dosage of VTX for maximum neuroprotective effects against OD?
Any drug can undergo metabolism and excretion. How does the intraperitoneal administration of VTX impact its bioavailability and efficacy? Is i.p the preferred administration route? Can oral administration provide the same effects?
Since the neuroprotective role of VTX has been shown, the authors should comment on the optimal dose and the probable sources of VTX in the diet that could provide such benefits.
The authors have clearly shown that VTX's neuroprotective mechanism, through the Nrf2 pathway, contribute to its protective effects against orofacial dyskinesia (OD). VTX significantly reduced HPD-induced OD as seen by VCM and TP, decreased oxidative stress(striatal Nitric Oxide (NO) and Lipid Peroxide Production Induced by HPD), enhanced antioxidant capacity (modulating the GSH, SOD and CAT levels), prevented mitochondrial dysfunction, and reduced neuroinflammatory and apoptotic markers in the striatum. Further, they showed that the protective effects of VTX were reversed on administration of trigonelline (TGN), an inhibitor of the Nrf2-mediated pathway.
Based on the results of OD in rates can the authors comment on the potential therapeutic benefits of Vitexin (VTX) in preventing or treating tardive dyskinesia (TD) in humans? What are the limitations of using the rat model to study OD and TD, and how can these findings be translated to human studies?
What are the authors views on VTX be used as a potential therapeutic candidate for the prevention or treatment of TD in clinical settings? Suggestions for future work
At line 257-265 authors describe the existing treatments for TD. How does VTX's neuroprotective effects compare to existing treatments for TD?
What may be the potential side effects or interactions of VTX with other medications in clinical use?
Add DOI for all references in the manuscript.
Comments on the Quality of English Languagegeneral English correction with reference to tense.
Author Response
Response to Reviewer 2 Comments
Dear Reviewers,
I’m writing in response to your feedback regarding the manuscript we submitted. Thank you so much for your positive comments and suggestions for the manuscript entitled. The manuscript has been revised based on your inquiries, rephrased the content of the manuscript and resubmitted through the journal website. The revised parts have been marked in red. The following is the response to your inquiries point-by-point; we hope our responses fully address your comments and suggestions:
Reviewers’ comments:
The paper entitled “Vitexin Mitigates Haloperidol-Induced Orofacial Dyskinesia in Rats Through Activation of the Nrf2 Pathway” has investigated the protective effects of Vitexin (VTX), a C-glycosylated flavone against orofacial dyskinesia (OD) in rats, induced by haloperidol (HPD), along with the neuroprotective mechanisms underlying these effects. Authors have performed the work with utmost scientific rigor and the work is commendable.
Response: Thank you so much for giving us such a detail constructive comments and suggestions.
Comments-
Point 1: In Figure 1- colored symbol for the data points may be useful for better visualization.
Response 1: Thank you so much for giving us such detail constructive comments and suggestions, we have changed these in the revised manuscript.
Point 2: In the discussion of results- the following details reproduced in the materials and methods at line 318- 320 may be included- “VTX was initially tested at 1 mg/kg with no sig-318 nificant effects, but significant results were observed at 10 and 30 mg/kg, particularly in 319 reducing HPD-induced OD. Therefore, 10 and 30 mg/kg were used in this study.” Based on the above results what is the optimal dosage of VTX for maximum neuroprotective effects against OD?
Response 2: Thank you for Thank you for your valuable suggestions, we have made the necessary revisions based on your feedback. This information has been added into the revised manuscript.
VTX was initially tested at 1 mg/kg with no significant effects, the dosage was incre-mentally increased to a maximum of 100 mg/kg to achieve statistically significant re-sults, but significant results were observed at 10, 30 and 100mg/kg, particularly in re-ducing HPD-induced OD. However, there were no significant variations at 30 and 100mg/kg. Therefore, 10 and 30 mg/kg were used in this study. According to these results, the optimal dosage of VTX for maximum neuroprotective effects against OD is 100mg/kg.
Point 3: Any drug can undergo metabolism and excretion. How does the intraperitoneal administration of VTX impact its bioavailability and efficacy? Is i.p the preferred administration route? Can oral administration provide the same effects?
Response 3: Thank you for your suggestions. Based on our current understanding, there are significant differences between intraperitoneal (i.p.) and oral administration in terms of drug delivery and efficacy. Intraperitoneal administration involves injecting the drug directly into the abdominal cavity, where it is rapidly absorbed through the abdominal blood vessels, often achieving systemic effects more quickly and with higher bioavailability, as it bypasses the digestive system. This method is suitable for situations requiring rapid absorption but may lead to abdominal inflammation or infection and is more invasive. In contrast, oral administration is the most common route, where the drug is absorbed through the digestive system, taking longer to produce effects and often with lower bioavailability due to potential degradation or metabolism in the digestive tract and the first-pass effect. Given these differences, we have chosen intraperitoneal administration for our study, hoping it addresses and answers your concerns.
Point 4: Since the neuroprotective role of VTX has been shown, the authors should comment on the optimal dose and the probable sources of VTX in the diet that could provide such benefits.
Response 4: Thank you for your suggestions. We have made the necessary revisions based on your feedback, and added into the revised manuscript. We hope these adjustments address your suggestions and appreciate how your input has enhanced the readability and completeness of the article.
Point 5: The authors have clearly shown that VTX's neuroprotective mechanism, through the Nrf2 pathway, contribute to its protective effects against orofacial dyskinesia (OD). VTX significantly reduced HPD-induced OD as seen by VCM and TP, decreased oxidative stress (striatal Nitric Oxide (NO) and Lipid Peroxide Production Induced by HPD), enhanced antioxidant capacity (modulating the GSH, SOD and CAT levels), prevented mitochondrial dysfunction, and reduced neuroinflammatory and apoptotic markers in the striatum. Further, they showed that the protective effects of VTX were reversed on administration of trigonelline (TGN), an inhibitor of the Nrf2-mediated pathway.
Based on the results of OD in rates can the authors comment on the potential therapeutic benefits of Vitexin (VTX) in preventing or treating tardive dyskinesia (TD) in humans? What are the limitations of using the rat model to study OD and TD, and how can these findings be translated to human studies?
Response 5: Thank you so much for giving us such a detail constructive comments and suggestions, regarding your questions, our responses are as follows. We hope these answers address your inquiries and provide the information you need. VTX has potential therapeutic benefits for preventing or treating tardive dyskinesia (TD) in humans, primarily due to its neuroprotective effects through the Nrf2 pathway. It can reduce oxidative stress, improve antioxidant capacity, and prevent mitochondrial dysfunction. Additionally, VTX can inhibit neuroinflammation and apoptotic markers, which may be beneficial for treating TD, a condition related to long-term neural damage. However, there are limitations to using rat models for studying orofacial dyskinesia (OD) and TD. Firstly, there are differences in neurophysiology and pathology between rats and humans, which may lead to discrepancies in drug efficacy. Secondly, OD as a rat model may not fully replicate the pathological processes of human TD, particularly its complex pathogenesis. Finally, differences in dosage and drug metabolism may also limit the translation of these findings to humans. To translate these findings to human studies, future clinical trials are needed to determine the efficacy and safety of VTX in TD patients. Further research is also necessary to establish the optimal dosage and long-term effects in humans. Additionally, developing animal models that more closely mimic human pathological features or utilizing other biotechnologies to better simulate the pathogenesis of human TD will help in more accurately assessing the potential therapeutic effects of VTX.
Point 6: What are the authors views on VTX be used as a potential therapeutic candidate for the prevention or treatment of TD in clinical settings? Suggestions for future work
Response 6: Thank you so much for the suggestion; Regarding your questions, our responses are as follows. We hope these answers address your inquiries and provide the information you need. This study provides strong evidence of Vitexin's (VTX) potential for treating orofacial dyskinesia (OD) in animal models. VTX appears to offer neuroprotection by reducing oxidative stress, preventing mitochondrial dysfunction, decreasing neuroinflammation, and inhibiting apoptosis, likely through interactions with the Nrf2 pathway, which is crucial for cellular defense. However further, cellular and molecular studies will be required to confirm the possible neuroprotective mechanisms of VTX. The use of HPD-induced OD in rats, which resembles human tardive dyskinesia (TD), establishes a solid foundation for further investigation. While VTX shows promise as an adjunct therapy for TD, translating these preclinical results into clinical practice requires additional research. Future clinical trials will be essential to confirm VTX's effectiveness and safety in humans, potentially improving treatment options for this debilitating condition. This information has been added into the revised manuscript.
Point 7: At line 257-265 authors describe the existing treatments for TD. How does VTX's neuroprotective effects compare to existing treatments for TD?
Response 7: Thank you. Regarding your questions, our responses are as follows. We hope these answers address your inquiries and provide the information you need. Existing treatments for tardive dyskinesia (TD) often involve antipsychotic and anticholinergic drugs, such as biperiden. However, these treatments can lead to side effects like tachycardia, mydriasis, dry mucous membranes, and urinary retention, and may also worsen positive symptoms of schizophrenia. This underscores the need for alternative therapeutic approaches. In contrast, Vitexin (VTX), a key bioactive compound from the traditional Chinese herb *Crataegus pinnatifida* (hawthorn), offers strong antioxidant, anti-inflammatory, and anti-apoptotic properties. VTX has been shown to prevent neurochemical deficits, provide neuroprotection, and effectively counteract orofacial dyskinesia induced by HPD. Thus, VTX may offer a promising alternative that avoids the side effects associated with traditional treatments, showing potential advantages in managing TD.
Point 8: What may be the potential side effects or interactions of VTX with other medications in clinical use?
Response 8: Thank you. Regarding your questions, our responses are as follows. We hope these answers address your inquiries and provide the information you need. Vitexin (VTX) may have several potential side effects and interactions with other medications in clinical use. Gastrointestinal discomfort, such as nausea or diarrhea, could occur in some individuals. Although rare, allergic reactions like rash or swelling may also happen. VTX might interact with other medications, particularly those metabolized by the liver, potentially altering their effectiveness, or increasing side effects. Additionally, VTX could have mild anticoagulant effects, raising the risk of bleeding when combine with other blood-thinning medications. When used with sedatives or central nervous system depressants, VTX might enhance their effects, leading to increased drowsiness. Its potential antidiabetic properties might also interact with diabetes medications, affecting blood sugar levels. Furthermore, VTX could influence the absorption of certain drugs, either enhancing or reducing their bioavailability. Therefore, it's important to monitor for any adverse effects or interactions and consult a healthcare provider before using VTX alongside other medications.
Point 9: Add DOI for all references in the manuscript.
Response 9: Thank you again, the references have been revised according to your suggestions.
Thank you for your valuable comments/suggestions and giving us the opportunity to revise the manuscript to a more readable level. We worked very hard to response your inquiries and to revise the manuscript. Before we finalized the revised manuscript, we have resent the manuscript for proof-reading as well as one last review by a native English-speaking researcher.We hope the manuscript could pass the review to be published in your prestigious journal: International Journal of Molecular Sciences
Sincerely yours,
Hsiang-Chien Tseng
Department of Anesthesiology, Shin Kong Wu Ho-Su Memorial Hospital,
No. 95, Wen Chang Road, Shih Lin District, Taipei 11101, Taiwan.
Fax: +886 2 833 2211.
E-mail address: [email protected] (Hsiang-Chien Tseng).
Round 2
Reviewer 1 Report
Comments and Suggestions for Authors
The manuscript can be accepted in the present form
Author Response
Thank you so much for the kind comment and wonderful suggestion.